# ReMod: Learning Structured Sparsity with ReLU Modulation

**Wenbo Zhang & Xiang Ren**
Department of Computer Science
University of Southern California
Los Angeles, CA 90007, USA
{wenboz,xiangren}@usc.edu

## Abstract

Large language models demand substantial computational resources for training and inference. Leveraging contextual sparsity to convert dense modules into sparsely computed Mixture of Experts (MoE) offers a promising solution, but existing methods face challenges in effectively partitioning modules and handling abrupt, non-differentiable changes during conversion. We introduce ReMod (ReLU Modulation), which creates sparsity smoothly and differentiably while integrating clustering directly into training. Our method trains a small ReLU-gated modulator that scales hidden states to sparsify computation, then clusters modulator weights to create structured sparsity with optimized hardware utilization. When applied to MLPs and Attention projections in Bert-base, ReMod reduces inference FLOPs by up to 93% while maintaining comparable accuracy—significantly outperforming previous approaches.

## 1 Introduction

Deep learning models, particularly transformers (Vaswani et al., 2017), have grown increasingly powerful in natural language processing and computer vision at the cost of substantial computational demands. As these models continue to scale, reducing their computational and memory requirements during both training and inference has become crucial. This paper specifically addresses inference computation costs, measured in FLOPs (or MACs) and wall-time latency.

One promising approach leverages contextual sparsity—the observation that only a subset of model components actively contribute to processing any given input (Liu et al., 2023). For practical implementation, this contextual sparsity must manifest as numeric sparsity, allowing computations to be skipped without affecting results, typically through zero values. A notable instance is activation sparsity in transformer architectures, where ReLU activations produce numerous zeros in MLP intermediate layers. Recent work (Zhang et al., 2022; Szatkowski et al., 2024; Lee et al., 2024) has demonstrated initial success converting standard MLP modules into sparsely computed Mixture of Experts (MoE) modules—a process termed MoEfication—by training routers to predict activation patterns across neuron clusters and skipping low-activation clusters. Building on this approach, (Zheng et al., 2024) proposed replacing prediction-based routing with a sparsity loss that encourages minimal routing scores.

For efficient GPU computation in MoEfication, neurons must be grouped into clusters (experts) that are processed or skipped as complete units. Previous methods typically accomplish this through balanced KMeans clustering on weights from the first linear layer in MLPs. Although Zhang et al. (2022) demonstrated superior performance using coactivation graph-based clustering, subsequent works avoided this approach due to its computational overhead. Developing more effective and efficient clustering techniques remains an important challenge. Additionally, the softmax or sigmoid routing mechanisms employed in previous works create abrupt, non-differentiable state changes when experts are skipped, complicating model adaptation to the modified architecture.

We introduce **ReLU Mod**ulation (ReMod), which creates sparsity through smooth, differentiable transformations while integrating clustering directly into the training process. ReMod trains a small modulator that produces ReLU-gated outputs to scale the main module's activations. To achieve

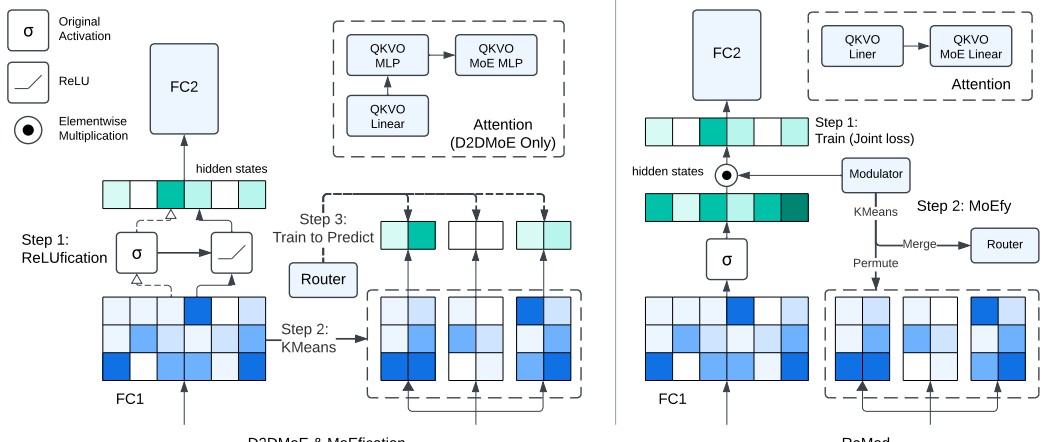

Figure 1: Comparison of ReMod with D2DMoE and MoEfication. Only some parts in the fine-tuning stage are shown. D2DMoE and MoEfication first replace the activation function and train the model to adapt to it, then cluster the first linear layer's weights to group them into experts, and finally train a router to predict their activation levels. ReMod instead trains a modulator that scales the hidden state to sparsify them and uses the modulator's weight as the feature for clustering to convert the modulator into a router. D2DMoE also replaces the QKVO projections in Attention with MLPs through distillation to MoEfy them, whereas ReMod can be directly applied to linear layers.

structured sparsity for optimal parallelization, we cluster the modulator weights to form an MoE router, using the resulting cluster labels to permute the main module's weights into cohesive expert groups. Our evaluation on Bert (Devlin et al., 2019) models shows that ReMod maintains classification accuracy comparable to dense models while reducing inference FLOPs by 90%—significantly improving upon D2DMoE's (Szatkowski et al., 2024) 62.6% reduction—while requiring over 99% less retraining.

## 2 ReLU Modulation

In this section, we introduce ReMod by demonstrating its application to linear layers as an exemplar (bias terms are omitted for simplicity but handled similarly). We'll discuss applications to other modules in subsequent sections. Throughout, we denote input size, hidden size, output size, and number of clusters as $i$, $h$, $o$, and $k$ respectively. Figure 1 contrasts ReMod with prior approaches (D2DMoE and MoEfication).

### 2.1 Goal

We begin by defining the target architecture produced by ReMod, as illustrated in Figure 2b. This represents the end goal of our training and MoEfication process, which we detail in subsequent sections.

Consider a module where features (output, hidden state, or input) can be partitioned into $k$ independently computable blocks. We refer to individual scalar elements of these features as neurons. The module is divided along the feature dimension into $k$ sub-modules (experts), each corresponding to one feature block. A compact, trained modulator (functioning as the router) produces $k$ scalar values, one per expert. These values, termed modulations, are predominantly zeros. Each feature subset is multiplied by its assigned modulation, allowing computation to be skipped for sub-modules with zero modulation.

For a concrete example, consider a linear module $\boldsymbol{y} = \boldsymbol{x}\boldsymbol{W}$ with weight matrix $\boldsymbol{W} \in \mathbb{R}^{i \times o}$. We permute and partition this matrix into $k$ experts of output size $\frac{o}{k}$, denoted as $\boldsymbol{W}_1$ to $\boldsymbol{W}_k$. For an input $\boldsymbol{x} \in \mathbb{R}^i$, the modulator generates an expert-level modulation $\boldsymbol{m} \in \mathbb{R}^k$ that assigns (mostly zero) scores to each expert. Each expert's output is scaled by its corresponding score, meaning only experts with non-zero scores require computation. The scaled outputs are concatenated (with

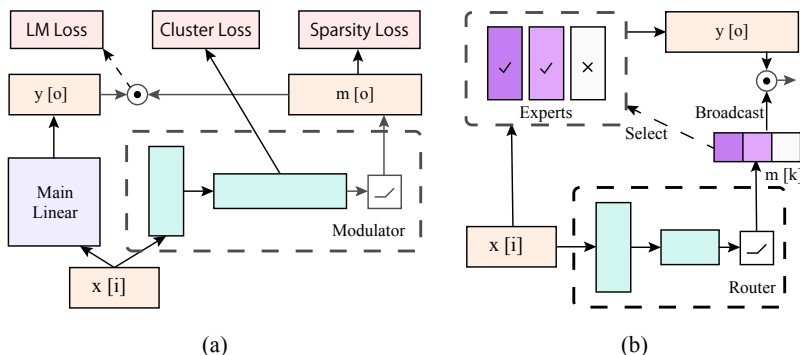

Figure 2: The forward method of a linear layer with ReMod during training (a) and inference (b).

zero-padding for inactive experts) and unpermuted to maintain the original output ordering:

$$\boldsymbol{m} = \text{Modulator}(\boldsymbol{x}), \quad \boldsymbol{y}_j = \begin{cases} \boldsymbol{m}_j \cdot \boldsymbol{x}\boldsymbol{W}_j & \text{if } \boldsymbol{m}_j > 0 \\ 0_{o/k} & \text{otherwise} \end{cases}, \quad \boldsymbol{y} = \text{unpermute}(\text{concat}(\boldsymbol{y}_0, \dots, \boldsymbol{y}_k))$$

To ensure the modulator remains significantly smaller than the linear module, we implement it as a two-layer MLP with a small hidden dimension $h \ll i, o$. This results in only $h(i + k)$ parameters in the modulator, compared to $i \times o$ parameters in the linear module. The MLP output passes through a ReLU activation to generate sparse modulations. Formally, $\text{Modulator}(x) = \text{ReLU}(\text{Act}(\boldsymbol{x}\boldsymbol{W}_d')\boldsymbol{C})$, where Act is an activation function (we use SiLU (Elfwing et al., 2018)), with $\boldsymbol{W}_d' \in \mathbb{R}^{i \times h}$ and $\boldsymbol{C} \in \mathbb{R}^{h \times k}$ as the MLP weights.

## 2.2 NEURON CLUSTERING

To effectively leverage sparsity on modern GPUs, we must address the challenge of structure. While individual neuron-level sparsity can be beneficial for small-scale inference (e.g., on-device decoding (Song et al., 2024)), its overhead becomes prohibitive at scale. The primary computational benefit emerges when we can skip entire groups of neurons—experts—rather than individual neurons.

The key insight of MoEfication techniques is to cluster neurons such that entire expert groups can remain inactive. Previous approaches (Szatkowski et al., 2024; Zhang et al., 2022) applied Balanced-KMeans clustering on the weights of first linear layers (or gate projections in GLU-activated models (Dauphin et al., 2017)), assuming neurons with similar weights would exhibit similar activation patterns. Our approach differs for two critical reasons: first, in ReMod, neuron activation depends on the modulation value rather than the neuron's intrinsic output; second, as we demonstrate in section 5.2, the correlation between weight similarity and activation patterns can be surprisingly weak.

Since neuron modulation patterns are not predetermined before training, we initialize the modulator to operate at neuron-level granularity and integrate clustering directly into the training process (illustrated in Figure 3). The modulator initially takes the form $\text{Modulator}(x) = \text{ReLU}(\text{Act}(\boldsymbol{x}\boldsymbol{W}_d')\boldsymbol{W}_u')$ where $\boldsymbol{W}_u' \in \mathbb{R}^{h \times o}$ is the output layer weight matrix with neuron-level granularity ($o$ outputs).

During each training step, we perform Balanced-KMeans clustering on $\boldsymbol{W}_u'$ to derive cluster centers $\boldsymbol{C} \in \mathbb{R}^{h \times k}$ and corresponding labels $\boldsymbol{l} \in \mathbb{N}^o$ (where $\boldsymbol{l}_j$ indicates which cluster the $j$th neuron belongs to). We then compute a cluster loss as the L1 distance between each neuron's weight vector and its assigned cluster center: $\sum_{j=1}^{o} ||W_{:,j} - C_{:,l_j}||_1$.

After training, we convert the neuron-level modulator into an expert-level router by replacing $\boldsymbol{W}_u'$ with the cluster centers $\boldsymbol{C}$ and using the cluster labels to reorganize the main module's weights into coherent expert groups. To ensure a seamless transition, we implement cluster mixing in the final training phase—gradually replacing each neuron's modulation with its cluster's mean modulation. This guarantees that the neuron-level and expert-level modulations produce identical outputs at inference time.

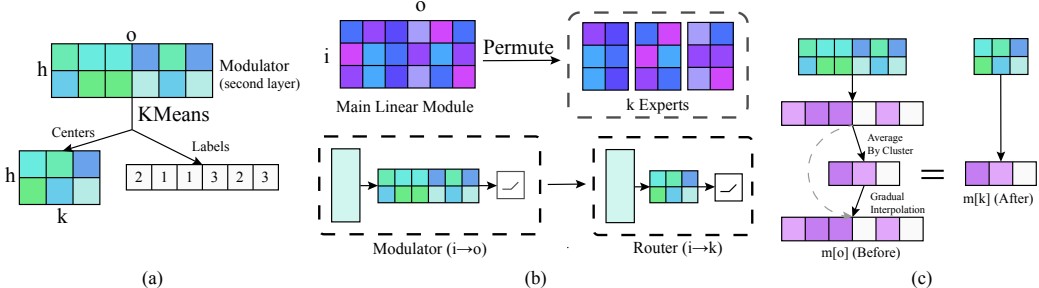

Figure 3: Illustration of the MoEfication process in ReMod: (a) in each training step, the weight matrix of the modulator's output layer $W \in \mathbb{R}^{h \times o}$ is used as features for balanced-KMeans clustering to obtain the cluster centers $C \in \mathbb{R}^{h \times k}$ and labels; (b) after training, the modulator is converted into a router by replacing $W$ with $C$, and the experts are constructed by permuting the weights of the main module using the labels; (c) near the end of the training, the neuron-level modulation is gradually replaced by the cluster average, so the router's output is identical to the modulator's.

## 2.3 SPARSIFICATION

To ensure a smooth transition, we initialize the modulator to function as an identity operation. We then progressively introduce sparsity through a dedicated loss term.

While L1 regularization is traditionally used for sparsification, it applies equal gradient magnitude to all values. Since our objective is to generate true zeros rather than merely small values, we employ a more targeted approach that applies stronger pressure to near-zero values while having less impact on larger values. Specifically, we implement an $L_p$ regularization loss normalized by output dimensionality: for a modulator output $m$,

$$\mathcal{L}_{\text{sparsity}} = \frac{1}{N} \sum_{i=1}^{N} |m_i|^p, \qquad \frac{\partial \mathcal{L}_{\text{sparsity}}}{\partial m_i} = \begin{cases} \frac{1}{N} p \cdot m_i^{p-1} & \text{if } m_i > 0 \\ 0 & \text{otherwise} \end{cases}$$

where $p \in (0, 1]$ is a hyperparameter (we use 0.5), and $N$ represents the neuron count. We implement gradient clipping and zero out gradients when $m_i = 0$.

## 2.4 JOINT FINE-TUNING

Sparsification, clustering, and router training are achieved in a single training cycle: we train the modulator to minimize the joint loss $\mathcal{L} = \mathcal{L}_{\text{task}} + \lambda_{sparsity}\mathcal{L}_{\text{sparsity}} + \lambda_{cluster}\mathcal{L}_{\text{cluster}}$ , where $\mathcal{L}_{\text{task}}$ is the loss for the target task (e.g., classification or language modeling), and $\lambda_{\text{sparsity}}$ and $\lambda_{\text{cluster}}$ are hyperparameters that control the importance of each objective. We implement a staged training schedule: initially training with only the task loss, then gradually introducing the sparsity loss, and finally incorporating the cluster loss.

# 3 APPLICATIONS

This section demonstrates the application of ReMod to different neural network modules, with additional design variants detailed in the appendix.

## 3.1 MLP

For a standard two-layer MLP, ReMod is applied to the hidden states—specifically the output of the first linear layer (or the product of down and gate projections in GLU-activated models (Dauphin et al., 2017)). This approach enables sparse computation in the second linear layer along its input dimension.

While it is theoretically possible to apply ReMod to the output of the second layer, enabling sparsity along both input and output dimensions, this approach introduces additional overhead from indexing

operations and CUDA kernel launches. Without optimized implementation and sufficiently large models, these overheads may outweigh the computational benefits of increased sparsity.

## 3.2 ATTENTION

We develop two ReMod variants for attention modules: projection modulation and attention score modulation. Although we consider the latter more promising, it lacks direct comparability with our current baselines. We therefore detail this approach in Appendix B and focus here on projection modulation.

Projection modulation converts the Query, Key, Value, and Output (QKVO) projections in attention modules into MoEs. Unlike D2DMoE, which requires replacing each projection with an MLP followed by distillation, ReMod can be directly applied to these linear layers. We also leverage an optional simplification: since Q, K, and V projections are naturally grouped by attention heads, we can designate each head as a cluster, initialize the modulators' output size to match the number of heads, and train without requiring the cluster loss.

## 4 EXPERIMENTS

### 4.1 SETUP

Generally, we followed the setup for evaluating Bert (Devlin et al., 2019) in D2DMoE.

**Datasets** We used the Carer emotion classification dataset (Saravia et al., 2018). Following Szatkowski et al. (2024), we padded all sequences to 128 tokens (we note that this sequence length is longer than all sequences in the dataset), and used the training split as both the training set and the validation split as the test set.[1]

**Model and Training Setup** We used the Adam optimizer to fine-tune Bert-base-uncased for 10 epochs with a batch size of 48. Figure 4 shows a comparison of training cost with D2DMoE (Szatkowski et al., 2024). We experimented with two granularity: expert size=24 and 128.

**Evaluation** We report the MACs and accuracy of our fine-tuned model. We compared our method with D2DMoE and MoEfication, the results of which are taken from Szatkowski et al. (2024). We used the same profiler as Szatkowski et al. (2024), namely the *fvcore* Flop Counter[2] [3]

### 4.2 RESULTS

As shown in Figure 4, ReMod achieved similar accuracy with significantly lower FLOPs compared to the baselines, and can further reduce the FLOPs at the cost of a slight decrease in accuracy (left). In addition, ReMod requires minimal retraining cost compared to D2DMoE and MoEfication (right).

### 4.3 WALL TIME SPEEDUP

We evaluated how well the theoretical FLOP reduction translates to wall time speedup. We used the same data as in the main experiments and measured the latency of the models on an A6000 GPU. To evaluate the importance of clustering neurons into experts, we included a setting where MLP and O projection modulators are not clustered.

Table 1: Wall time speedup

| Model | GMACs | Samples/s |
|---|---|---|
| Dense | 11.20 | 735 |
| ReMod | 0.75 | 1809 |
| ReMod (N) | 1.184 | 140 |

As shown in Table 1, ReMod achieved a $2.46\times$ wall time speedup compared to the dense model. The model without clustering (named as ReMod (N) in the table) suffered from immense

---

[1] https://github.com/bartwojcik/D2DMoE/blob/0d80c763e5/data_utils/data.py#L788-L827

[2] https://github.com/facebookresearch/fvcore

[3] This profiler shows *FLOPs* but actually reports *MACs*, where $1\text{MAC} \approx 2\text{FLOPs}$.

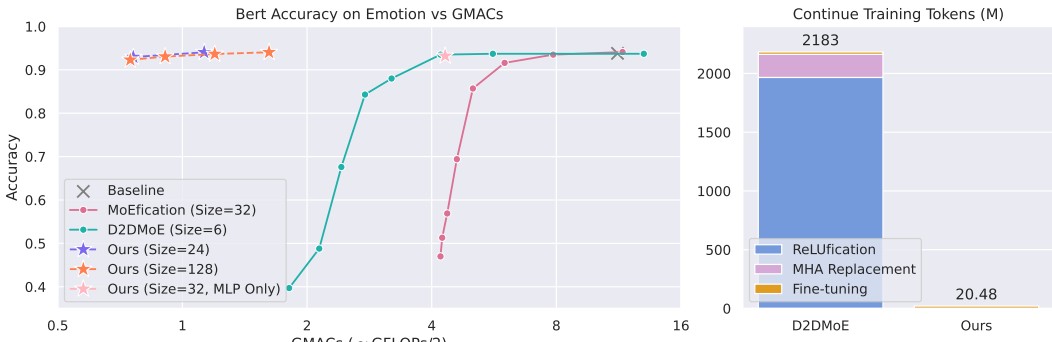

Figure 4: Comparison of ReMod with D2DMoE and MoEfication. ReMod achieved comparable accuracy with significantly lower FLOPs even with larger expert size (left), while also requiring minimal continue training cost (right). We also report the result of our method without modifying the attention projections (MLP Only).

overheads, which caused an 80% slowdown compared to the dense model despite having only 10.6% of the FLOPs, which is also larger than with clustering due to the larger modulator size.

Although the speedup was evident, it did not fully align with the theoretical gains implied by the MAC reduction. Profiling with the PyTorch Profiler revealed that, after applying ReMod, matrix multiplications in MLP and QKVO projections, combined with the modulators' computation, occupied only about 23.6% of each layer's runtime. In contrast, indexing operations accounted for 18.9%, the remaining attention components comprised 14.8%, and the remaining 42.6% falls into Layer Norm and miscellaneous overheads beyond the scope of this work (details in Appendix A). We anticipate these overheads to become a smaller fraction of the total runtime in larger models, thus delivering more pronounced speedups in practice.

## 5 ANALYSIS

### 5.1 EFFECT OF LOSS TERMS

We evaluated the effect of the sparsity and cluster loss through ablation studies. The same dataset was used, except that the trivial sparsity caused by the pad tokens was excluded from the results. To better match realistic application scenarios where it is important to parallelize computation, the large expert size setting (24 experts of size 128 in MLP, 12 experts of size 64 in each attention projection) was used.

For our full method, we included two settings: mild ($\lambda_{sparsity} = 1$) and aggressive ($\lambda_{sparsity} = 8$). The results were compared to a model trained without the sparsity loss ($\lambda_{sparsity} = 0$), and models trained without the cluster loss in the aggressive setting. In the latter, we still used the same clustering algorithm and gradual replacement of neuron-level modulation by cluster-level modulation, but set $\lambda_{cluster}$ to 0. For the aggressive setting and the no-cluster-loss setting, we repeated the experiment 4 times with batch sizes 46, 48, 49, and 50, and report the mean and standard deviation of the accuracy. As shown in Table 2, the sparsity loss significantly increased the sparsity while

| Model | Accuracy | MLP% | QKVO% |
|---|---|---|---|
| No Sparsity | 94.00 | 79.73 | 93.91 |
| ReMod (Mild) | 94.05 | 62.65 | 48.42 |
| ReMod (Aggressive) | 92.20±0.13 | 22.36 | 11.98 |
| No Cluster (Aggressive) | 90.57±1.07 | 16.64 | 9.79 |

Table 2: Results of the ablation study. MLP% and QKVO% indicate the percentage of the MLP and QKVO projections being activated (i.e., requires computation).

maintaining the performance in the mild setting. Removing the cluster loss made the training less

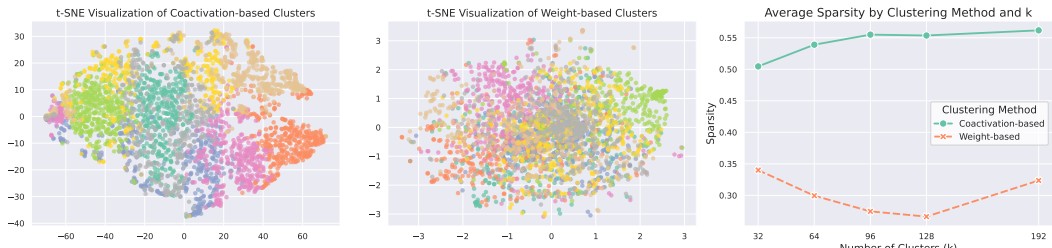

Figure 5: Left and middle: visual comparison of clustering with coactivation vs. weight (k=8). Right: comparison of resulting sparsity of two methods with various k; a cluster is considered inactive when $< 4\%$ of its neurons are activated.

stable, as depicted by the lower mean and larger standard deviation in accuracy. Intuitively, while the model can learn to adapt to the gradual merger of the clusters, without the clustering loss the weights in each cluster do not become similar to each other, and thus are more likely to be abruptly assigned to different clusters in the next iteration, making the training less stable.

## 5.2 CLUSTERING METHODS

Zhang et al. (2022) compared the result of clustering using the first linear module's weight as the feature versus using the coactivation graph, and found the latter to have better overall performance. However, collecting and processing coactivation graphs requires extra resources, so the former was adopted in subsequent works.

The use of weights as the clustering feature is based on the assumption that similar weights will have similar activation patterns. In contrast, the coactivation graph directly captures the true activation pattern. To test whether the assumption is true, we compare the clustering results of the two methods. We define the coactivation matrix $K$ where each element $K_{i,j}$ represents the frequency with which neurons $i$ and $j$ are simultaneously activated given the same input. To compare clustering methods, we collected activation statistics from the wikitext dataset (Merity et al., 2016) using the ReLUfied Bert model provided by Szatkowski et al. (2024). We then performed clustering using both the coactivation matrix and weight matrix as features, and visualized the results using t-SNE with $K$ as the feature space for both methods. As illustrated in Figure 5 (left and middle), weight-based clustering demonstrates significantly inferior clustering quality compared to coactivation-based clustering. This quality difference directly translates to practical performance: Figure 5 (right) shows that coactivation-based clustering achieves nearly twice the sparsity of weight-based clustering across various cluster counts.

Our analysis revealed two fundamental limitations of weight-based clustering. First, without appropriate regularization, neuron weights exhibit minimal natural similarity in high-dimensional spaces—in ReLUfied Bert, the average distance between a neuron and its nearest neighbor (1.21, measured by L2 norm) exceeds the average weight norm itself (1.09). Second, the binary nature of activation (where neurons abruptly switch at the threshold) means that even neurons with similar output values may have completely different activation states, further weakening the weight-activation pattern correlation. ReMod effectively addresses both issues: the cluster loss explicitly encourages weight similarity within clusters, while the continuous modulation values ensure that similar modulator weights directly translate to similar activation patterns, creating a more reliable clustering foundation.

## 6 DISCUSSION

In this work, we introduced ReMod, a novel approach for efficient sparsification and MoEfication of dense neural networks. Our experiments with Bert demonstrate that ReMod maintains comparable accuracy while substantially reducing computational requirements—achieving both theoretical FLOP reductions and practical wall time speedups. Notably, ReMod eliminates the dependency on pre-existing sparsity patterns in original modules, thereby significantly reducing retraining costs compared to previous methods like D2DMoE and MoEfication.

While our current evaluation has focused primarily on Bert models with limited comparative baselines, we intend to expand our experimental scope substantially. Future work will investigate ReMod's effectiveness across a broader spectrum of architectures, including decoder-only models and non-transformer networks. Additionally, we plan to explore potential applications in KV-Cache sparsification and sequence compression, as outlined in the appendix.

It is worth noting that recent work by Wang et al. (2025) demonstrated that pretraining MoE models with ReLU routing (termed ReMoE) yields performance improvements. The architecture resulting from applying our method to a MLP is structurally identical to a ReMoE MLP.

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

## A    INFERENCE LATENCY BREAKDOWN

See figure 6.

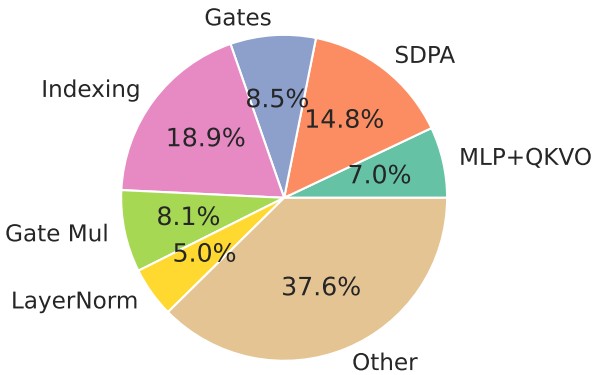

Figure 6: Latency breakdown

## B    ANOTHER VARIANT OF ATTENTION MODULATION

In this section, we introduce another variant of ReMod for the Attention module, which applies modulation on the attention scores. This variant aligns with the intuition that each attention head

only assign meaningful attention scores to a subset of tokens, and the remaining, small attention values can be sparsified. It also has the potential to reduce VRAM consumption by compressing the KV-Cache.

A single Attention head in a classical Multi-Head Attention module (Vaswani et al., 2017) process an input $\boldsymbol{X} \in \mathbb{R}^{l,h}$ by

$$\boldsymbol{Q} = \boldsymbol{X}\boldsymbol{W}_Q, \ \boldsymbol{K} = \boldsymbol{X}\boldsymbol{W}_K, \ \boldsymbol{V} = \boldsymbol{X}\boldsymbol{W}_V \tag{1}$$

$$\boldsymbol{Z} = \frac{\boldsymbol{Q}\boldsymbol{K}^T}{\sqrt{d_k}} \quad (\boldsymbol{Z} \in \mathbb{R}^{l,l}) \tag{2}$$

$$\text{Attention}(\boldsymbol{Q}, \boldsymbol{K}, \boldsymbol{V}) = \text{softmax}(\boldsymbol{L})\boldsymbol{V},$$

$$\text{where softmax}(\boldsymbol{L})_i = \frac{\exp(\boldsymbol{Z}_i)}{\sum \exp(\boldsymbol{Z}_i)} \tag{3}$$

We apply three modulations on Attention: Q, KV, and matrix modulation. For a token and attention head, the Q and KV modulations adjust entire rows and columns of the attention matrix, respectively. Essentially, when the Q modulation is 0 for a Query head, the token cannot attend to any token through that head; when the KV modulation is 0 for a pair of Key-Value heads, the token cannot be attended by any token on that head. Thus, the corresponding computation of QKV projections (equation 1) and KV-cache can be discarded. Notice that the QKV projections are already grouped by heads, so clustering is not needed and we directly initialize the modulators' output dimension to match the number of heads, then train with only the task loss and the sparsity loss. For a model with $n_Q$ Query heads and $n_K$ Key-Value heads, the Q and KV modulations are computed as follows:

$$\boldsymbol{M}_Q \in \mathbb{R}^{l,n_Q} = \boldsymbol{X}\boldsymbol{W}_{Q_M} + \boldsymbol{b}_Q$$
$$\boldsymbol{M}_K \in \mathbb{R}^{l,n_K} = \boldsymbol{X}\boldsymbol{W}_{K_M} + \boldsymbol{b}_K$$

The matrix modulation operates on a more granular level, adjusting the attention matrix element-wise. For clarity, we discuss how it is applied to a single attention head (except for the Q and KV modulators, which directly outputs a score for each head); the same process is repeated for all heads. The modulator consists of an extra pair of (small) Q and K heads, whose outputs are multiplied to produce attention logits of the same shape as $\boldsymbol{Z}$ by equation 2:

$$\boldsymbol{Q}_M = \boldsymbol{X}\boldsymbol{W}_{Q_M}, \ \boldsymbol{K}_M = \boldsymbol{X}\boldsymbol{W}_{K_M}$$
$$\boldsymbol{M}_{mat} = \text{ReLU}(\frac{\boldsymbol{Q}_M\boldsymbol{K}_M^T}{\sqrt{d_k}})$$

This added granularity allows higher sparsity in the attention matrix, and therefore reduces the cost of computing the logits (equation 2). The logits are then passed through a ReLU function instead of softmax.

The outputs of the three modulators are merged by broadcasting the Q and KV modulations and then multiplying with the attention matrix:

$$\boldsymbol{M} = \boldsymbol{M}_{mat} \odot (\boldsymbol{M}_Q)_{:,\text{newdim},m} \odot (\boldsymbol{M}_K)_{\text{newdim},:,m}$$

Where $m$ is the index of the current attention head, newdim denotes inserting a new dimension of size 1 and broadcasting. Then, the modulation is applied during the softmax operation. Specifically, we replace the original softmax function (equation 3) with the modulated softmax:

$$\text{MSoftmax}(\boldsymbol{Z}, \boldsymbol{M})_i = \frac{\boldsymbol{M}_i^2 \odot \exp(\boldsymbol{Z}_i)}{\sum \boldsymbol{M}_i \odot \exp(\boldsymbol{Z}_i) + \epsilon}$$

where $M$ is the output of the matrix modulation, $\epsilon$ is a small value for avoiding zero division. This function is designed to satisfy the following properties:

- $\text{MSoftmax}(\boldsymbol{Z}, 1_{l \times l}) = \text{softmax}(\boldsymbol{Z})$;
- $\text{MSoftmax}(\boldsymbol{Z}, 0_{l \times l}) = 0_{l \times l}$;
- $\text{MSoftmax}(\boldsymbol{Z}, \boldsymbol{M})$ is stable (no abrupt changes) as $\boldsymbol{M} \to 0_{l \times l}$

## C    APPLICATION IN SEQUENCE COMPRESSION

Autoencoder is a common choice for unsupervised learning. Its goal is to compress the input features into a lower-dimensional latent representation, by learning a pair of encoder and decoder that compress and reconstruct the input respectively so that the reconstruction loss is minimized. However, this latent representation has a fixed shape, which is suboptimal when the amount of information in the input has a large variance across samples.

For example, byte-level language modeling (Pagnoni et al., 2024; Wang et al., 2024) and latent space language modeling (team et al., 2024) both involve converting a long, low-level sequence into a shorter sequence of high-level latent patches to allow for efficient processing. However, deciding the size and position of each patch is non-trivial. A potential solution is to use an autoencoder that consists of two sequence models (e.g. transformers) and apply ReMod on the encoded sequence to induce token level sparsity (i.e., mask out entire tokens). Then, the non-zero token embeddings in the resulting sequence can be used as input embeddings to a large, latent-space language model.

