# OpenReview forum: "ReMod: Learning Structured Sparsity with ReLU Modulation"
_ICLR.cc/2025/Workshop/MCDC — MCDC @ ICLR 2025_

### Official Review · Reviewer_4qoA · 2025-02-23

**Rating:** 5
**Confidence:** 5
**Fit:** 4

**Summary:**

This paper presents ReLU Modulation as a novel approach to introduce sparsity and convert dense neural network modules into Mixture of Experts (MoE) post hoc. Unlike existing Moefication methods, which rely on predicting activation sparsity within MLP modules, ReM directly trains a modulator gated by ReLU to sparsify hidden states. This approach generalizes beyond MLPs, applies to linear layers and attention mechanisms, and achieves significant FLOP reductions (up to 93% in inference) while maintaining accuracy with minimal retraining cost.

**Reason For Giving A Higher Score:**

NA

**Reason For Giving A Lower Score:**

1. Promising method, but evaluation is too limited.
2. Practical speedup does not fully match theoretical FLOP reduction.
3. Needs comparisons with other sparsification techniques.
4. Would be stronger if evaluated on more model architectures (e.g., decoder models, CNNs).

**Strengths And Weaknesses:**

Pros of the paper:
1. Unlike existing Moefication techniques that rely on predicting activation sparsity, ReM directly introduces sparsity through a modulator trained alongside the model.This eliminates the need for ReLUfication of non-ReLU models, making it applicable to a broader range of architectures.

2. Prior Moefication methods were mainly applicable to MLPs since they depended on activation sparsity in feedforward layers. ReM removes this limitation and is successfully applied to linear layers and attention mechanisms,making it more versatile.

3. Achieves up to 93% FLOP reduction in inference, which is significantly higher than previous MoEfication methods like D2DMoE (62.6% FLOP reduction). Also, instead of sparsifying at the neuron level (which could introduce irregular sparsity), ReM clusters neurons into groups. This makes it easier to convert the dense model into a structured MoE, which is better suited for parallel execution on GPUs

Areas for improvement:
1. The paper introduces a sparsity loss using an Lp norm with p=0.5, but no theoretical explanation is provided on why this particular value was chosen.

2. All experiments are only conducted on BERT-base. The method should be tested on decoder-only models (GPT-style transformers) or vision models (ResNet, Swin Transformer) to confirm generalizability. It’s unclear how well ReM would work for models with highly structured activation patterns.

3. Theoretically, 93% FLOP reduction should result in more speedup, but the actual speedup is only 2.46x in wall time. The paper attributes this to indexing overhead, attention computations, and layer normalization costs, but further optimization could be done to improve practical gains.

4. The paper should compare ReM not only against the dense BERT-base model but also against other optimization techniques beyond just Moefication methods like D2DMoE. Moefication is not the only way to achieve efficiency. Without these baselines, it’s hard to fully assess whether ReM is the best choice for optimizing inference efficiency.

**Suggestions:**

Mentioned in the above section

---

### Official Review · Reviewer_4HsL · 2025-02-28

**Rating:** 6
**Confidence:** 2
**Fit:** 4

**Summary:**

The paper proposes ReM, which aims to MoEfiy the model, to achieve better computational efficiency. Compared to previous approaches that have different components for MoEfications and RELUfication, ReM has one router training that includes all the steps. Experiments are BERT model showing that the method can achieve competitive performance while using fewer FLOPs.

**Reason For Giving A Higher Score:**

ReM achieves strong token efficiency and the computation efficiency of the approach on BERT models.

**Reason For Giving A Lower Score:**

Currently, I have some clarification questions about the details. Also, the experimental setups are obsolete.

**Strengths And Weaknesses:**

[Strength]
1. The experimental results on BERT show the token efficiency and the computation efficiency of the approach.

[Weakness]
1. In Iine 116, the paper writes "D2DMoE had to replace the attention projections with MLPs to convert them into MoE". I assume the attention projections mean the `v_proj` and `o_proj` (value projection and output projection) in transformer architecture, and aren't they already forming as MLP layers?

2. I currently don't get why the paper uses clustering to cluster and merge the weights in modulators. My understanding of using clustering in previous approaches is that: we want to group the weights that are often activated together for the inputs so that we use clustering to find the groups and then only need to use one group, which is a subset of weights (denoting experts), at one time. The only thing we need is to label from clustering. But this paper also merges the modulator's weight based on clustering results, and I am not sure what is the purpose and motivation of it. Won't this merging make the modulator's prediction more inaccurate?

**Suggestions:**

* It would be better to introduce the architecture and its function for the modulator more clearly before Sec 2.1 and Sec 2.2. The paper currently has no clear definition of the architecture of the modulator until Sec 3.1, so I was confused about the meaning of the output layer of the modulator in line 145 in Sec 2.2.
* The current experimental setup is a little bit obsolete. It would be great if the paper could apply the approach to more recent language models and tasks.

---

### Official Review · Reviewer_TDfP · 2025-02-28

**Rating:** 6
**Confidence:** 5
**Fit:** 2

**Summary:**

The paper develops a new sparsification technique for models which involves using fine-tuned and clustered ReLU modules to induce sparsity patterns that can be well accelerated on modern hardware, such as GPUs.

**Reason For Giving A Higher Score:**

I think it is a neat and simple idea which might foster some discussion at the workshop.

**Reason For Giving A Lower Score:**

I think currently the empirical evidence for this method is to thin to discuss it meaningfully. The idea itself is valuable, but more experiments would need to be run to discuss if this is a method that other researchers can build on. I think for a workshop this is okay though.

**Strengths And Weaknesses:**

Strengths:
The idea is simple and compelling.

Weaknesses:
The experimental setup is on just one dataset that is relatively uncommonly used and only for a BERT model. The benchmark stems from some previous work. In itself, this result makes it difficult to establish if this method works or not. There are too many confounding factors to say how well this method does.

**Suggestions:**

For BERT in particular running on the RoBERTa fine-tuning suite is a good start, but BERT models are known to be easily compressible. I would also suggest to try this technique on LLMs. The baseline for other sparsification papers often demonstrate good quality compression and sparsification ratios on models like Llama 2 or 3 while using the LLM eval harness to get results for various datasets. I would recommend going this approach to verify the authors novel method.

---

### Decision · Program_Chairs · 2025-03-06

**Decision:**

Accept

**Comment:**

This paper proposes a new moefication/sparsification method to improve computational efficiency. The paper has been positively received by the reviewers who found enough contribution for acceptance at the workshop. We encourage the authors to take reviewers' comments and suggestions into consideration for the final version of the paper.